behaviour/cognition/evolution

facial expression, guilt, emotion, friendship, social consequences, morality

**Author for correspondence:**
Eglantine Julle-Danière
e-mail: eglantine.julle-daniere@port.ac.uk

# The social function of the feeling and expression of guilt

Eglantine Julle-Danière[1], Jamie Whitehouse[2],
Aldert Vrij[1], Erik Gustafsson[1] and Bridget M. Waller[2]

[1]Department of Psychology, University of Portsmouth, Portsmouth, Hampshire PO1 2UP, UK
[2]Department of Psychology, Nottingham Trent University, Nottingham, Nottinghamshire NG1 4FQ, UK

 EJ-D, 0000-0002-6052-1073; BMW, 0000-0001-6303-7458

Humans are uniquely cooperative and form crucial short- and long-term social bonds between individuals that ultimately shape human societies. The need for such intense cooperation may have provided a particularly powerful selection pressure on the emotional and communicative behaviours regulating cooperative processes, such as guilt. Guilt is a social, other-oriented moral emotion that promotes relationship repair and pro-sociality. For example, people can be more lenient towards wrongdoers who display guilt than towards those who do not. Here, we examined the social consequences of guilt in a novel experimental setting with pairs of friends differing in relationship quality. Pairs of participants took part in a cooperative game with a mutual goal. We then induced guilt in one of the participants and informed the other participant of their partner's wrongdoing. We examined the outcome using a dictator game to see how they split a joint reward. We found that guilty people were motivated to repair wrongdoing regardless of friendship. Observing guilt in others led to a punishment effect and a victim of wrongdoing punished close friends who appeared guilty more so than acquaintances. We suggest, therefore, that guilt has a stronger function between close friends as the costs of relationship breakdown are greater. Relationship context, therefore, is crucial to the functional relevance of moral emotions.

## 1. Friendship modulates the punishment function of guilt

### 1.1. Background

Humans are uniquely cooperative [1], and form crucial short- and long-term relationships between individuals (for review, see [2]). Such social bonds have proven to be adaptive across species,

providing an increase in fitness both of the individuals involved [3,4], as well as of their relatives, and their role in shaping human societies seems particularly pronounced [3]. The need for intense cooperation may have provided a particularly powerful selection pressure on the many emotional and communicative behaviours regulating cooperative processes. Such relationships involve cooperative interactions widely separated in time and space (reciprocal, and even delayed, altruism [5,6]), based on the memory of past interactions and the emotional load associated with them [2,4]. The costs associated with delayed altruism are higher than those associated with direct cooperation (there is a risk that the partner will not return the favour [7,8]), necessitating the simultaneous evolution of a control system. Punishment and spite have the potential to maintain cooperative behaviour and ensure equity within the relationship [9–11] by imposing costs on defecting partners. The costs could be refusal to cooperate at a later time or damage to the defecting partner's reputation at the scale of the whole social group [12], leading to wider repercussions against the untrustworthy. To regain the good graces of their partners, defectors need to acknowledge their wrongs, make amends and even express remorse regarding their wrongdoing [13]. Moral emotions are, therefore, intimately linked to our relationships with others, [14], facilitating the social interactions and important relationships [15,16].

Guilt is a social, other-oriented emotion that people experience regularly throughout life [17–19]. Evidence suggests that guilt has a potentially positive function within social interaction by stimulating pro-social behaviours from others, and also promoting actions towards those who have been wronged [18,20,21]. As the quality of relationship and likelihood of repeated interaction between two individuals is likely to affect the consequences of social transgressions [13], however, guilt may perform different roles depending on friendship status. For example, guilt could have a stronger impact within existing friendships compared with new or less close friendships [22,23], as the costs of relationship breakdown are greater. Regardless of any proximate penalties (e.g. punishment), there could be long-term benefits to both parties if the social bond is reinforced. It could thus be advantageous to communicate a feeling of guilt unambiguously within social interaction [18,24–26], given that expressing such feelings could maintain the social bond long-term.

Cooperation is a widespread strategy in humans, but some cultural differences have previously been identified [27–29]. In collectivistic settings, people develop strong bonds and cooperate mostly with close relatives, investing time and resources in the relationship [29]. In such cultural settings, it is possible that individuals have a clearer signal of guilt to reduce any potential punishment. In line with this hypothesis, cultural differences in the experience [30] and production/perception of emotion [31] have been demonstrated by recent work. However, these processes do not seem fixed. Recent work has shown that people changing cultural environment can 'adapt' to their new culture by experiencing emotional acculturation, slowly beginning to experience and express emotion according to their new cultural standards [32]. Those cultural variations emphasize the need for culturally and geographically varied samples to capture the general, and potentially universal, responses and identify a social function of guilt [28].

## 1.2. Present investigation

In an experimental study, pairs of friends and pairs of strangers took part in a cooperative task. We then artificially induced guilt in one participant by attributing failure in the task to this person, and informed the other participant of their partner's wrongdoing. We examined the influence of friendship, facial expression and culture on the participants' responses using a dictator game [33]. We decided to use the dictator game as a realistic, ecologically valid, measure of cooperation. The participants were asked to split the reward between their partner and themselves, however they wished to, and were told that this would be the amount each player would receive as payment. Our study is the first one, to the best of our knowledge, to examine the function of guilt between interacting participants, rather than via a hypothetical game with online partner or autobiographical recall. We aimed to replicate previous findings showing the pro-social function of guilt and test whether friendship affects how wrongdoers who are perceived as guilty are treated.

# 2. Methods

## 2.1. Participants

The sample size for this study was based on previous research looking at the induction of guilt. Moreover, due to financial limitation, and to keep the design as simple as possible, we did not

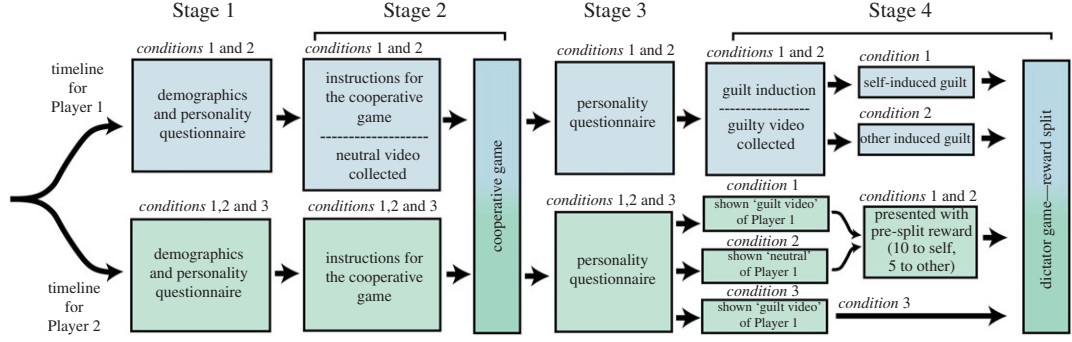

**Figure 1.** General procedure. A flowchart representing the procedure of the experiment.

include a 'no guilt' condition for Player 1. This allowed us to minimize the number of variables, already extensive, and reduce the number of participants needed. Therefore, in order to capture as many facial expressions as we could in a culturally varied sample, we aimed to have 50–60 participants posing as Player 1 in each condition (two conditions for Player 1: 'blame' versus 'no blame'). As a result, we aimed to collect data from 110 pairs of participants.

Two hundred and eighteen participants were recruited as pairs (109 pairs in total: 107 as Player 1 and 108 as Player 2, with three participants excluded). Participants were given the opportunity to either sign up with one of their friends or sign up alone and be randomly paired up with a stranger (36 participants signed up alone to be opportunistically paired up with someone else). Participants had various ethnicities and nationalities, constituting a sample made of individuals with various places of origin (PoO) (see electronic supplementary material, table S1 for details [28]). The origins of participants were clustered into two regions: Europe and East Asia. One hundred and twenty-eight participants had European nationality (81 female; mean age = 25.30, s.d. = 8.35) and 87 participants had East-Asian nationality (including some dual ethnicities; 55 female; mean age = 23.44, s.d. = 5.46; see electronic supplementary material, table S1).

## 2.2. Procedure

Participants were randomly attributed a role (Player 1 or Player 2; see figure 1). In **Stage 1**, participants were separated to complete questionnaires ('How am I in general' [34], guilt and shame proneness scale (GASP) [35], mood check (positive and negative affect schedule; PANAS) [36] and the unidimensional relationship closeness scale (URCS) [37]; see electronic supplementary material for details). The cooperative task (figure 2) was then explained and a video was taken of Player 1 during this time, with focus on the face (this video would act as a control, non-guilty stimuli at a later stage for Player 2—Stage 4). Participants were then reunited for **Stage 2**: a cooperative game where the participants had to work together to move a device from one end of the table to the other without dropping a marble (inspired from [38]; see figure 2). Participants were allocated at the beginning of the game a shared reward of £20 and were informed that the shared reward would reduce each time the marble was dropped. On average, the marble was dropped (i.e. a task failure) 10.32 times per pair (s.d. = 3.95; min: 3, max: 22). Due to the nature of the task, the attribution of responsibility remained largely ambiguous throughout and participants were not able to determine whose fault it was as the game went on. On completion of the task, the participants were separated again for **Stage 3**, where they completed more questionnaires (mood check, a friendship closeness scale, and the dirty dozen (DD) [39]). In **Stage 4**, both participants were then randomly allocated into different conditions (see below) and completed one final mood check before being asked to split the reward between their partner and themselves (see electronic supplementary material for the detailed procedure).

### 2.2.1. Stage 4—Player 1

Player 1 was informed that his/her individual performance on the task was responsible for their failure as a team and that the shared reward would be £15 instead of £20. The experimenter explained that we were monitoring concentration and motor abilities through observational analysis and that his/her concentration and motor coordination were lower than their partner's. During this feedback session, the face and upper body of Player 1 was video recorded (this video would act as an experimental,

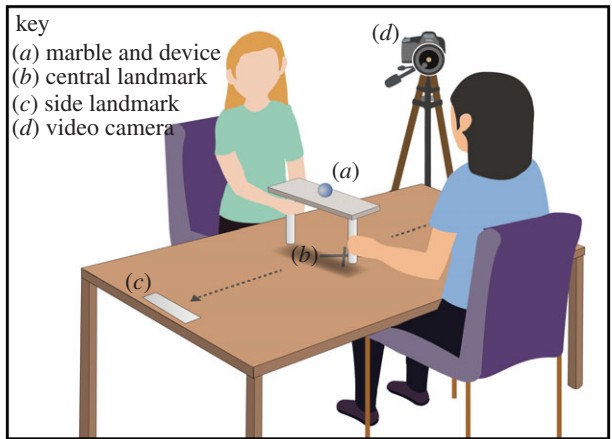

**Figure 2.** Cooperative game. For the cooperative game, two individuals were required to balance a marble on top of a single device that required two people to operate. They were instructed that the aim of the task was to see how many times the device (with balancing marble) could be moved from one end of a table to another, without the marble falling within 3 min. If/when the marble falls—the participants would repeat the task again until the time was up.

guilty stimuli for Player 2—Stage 4). The player was asked to split the reward (£15) between themselves and their partner (i.e. a dictator game [33]). This money split was designed as a measure of relationship repair, potentially to mend the previous transgression. Player 1 was then randomly allocated into a 'blamed' condition (told that Player 2 understands that it is their fault that they failed and thinks it would be unfair to split the money equally: other-induced guilt) or not blamed (told nothing: self-induced guilt). Due to financial limitation, and to keep the design as simple as possible, we did not manipulate whether Player 1 had reason to feel guilty. This allowed us to minimize the number of variables, already extensive, and reduce the number of participants needed.

### 2.2.2. Stage 4—Player 2

Player 2 was allocated to one of three conditions. In condition 1, they were told that the £20 reward was reduced to £15 due to the poor performance of Player 1, but that Player 1 had already split the £15 reward in Player 1's favour (10 for Player 1/5 for Player 2). This unfair original split was designed as a social transgression and an injustice. We wanted to measure Player 2's propensity to rectify a social injustice by assessing the change made to this original split. They were then presented with a 'guilty' video of Player 1 (when Player 1 received the poor feedback from the game) and were offered the chance to change the reward split (dictator game [33]). Condition 2 was identical but the video of Player 1 was 'neutral' (video taken when the game was explained in Stage 1). An additional control condition (condition 3) included the 'guilty' video but in the absence of any contextual information (not told that it was Player 1's fault and not told that Player 1 had pre-split the reward; both participants shared the failure). In all conditions, Player 2 was asked to judge the emotional state of Player 1 (including how guilty they thought Player 1 looked in the video).

## 2.3. Experimental design

Participants acting as Player 1 took part in a between-subject design, looking at the effect of blamed versus not blamed on reward split. Participants acting as Player 2 took part in a between-subject design, looking at the effect of guilty video versus control video on reward split. The dependent variable was the division of monetary reward. For Player 2, there was an additional control condition designed as a baseline to check the role of a potential guilt expression in the absence of context.

## 2.4. Measures

### 2.4.1. Facial expression

The stimulus videos of Player 1 were coded using the facial action coding system (FACS [40,41], see electronic supplementary material for details). Here, following the approach used in a previous

experiment [24], we used a bootstrapping approach to test the overall difference between the control videos and the guilt videos to see whether there were differences in the facial expressions between experimental interventions. This allowed us to control for a number of non-independent factors (e.g. culture, inter-individual differences). We also tested the differences between the videos judged as guilty by Player 2 and the videos not judged as displaying guilt in Player 1. In all analyses, we controlled for participants' PoO and level of friendship of the pair (see electronic supplementary material). Analyses were conducted in R v. 3.6.1 [42].

### 2.4.2. Dictator game—reward split

In the dictator game presented to Player 1 (figure 1), players were forced to make an uneven split and attribute at least £1 more to themselves or to their partner. Therefore, the dependent variable was the overall proportion of coins given to Player 2 (of the 15 available: *percentage* less than 0.5: more money retained for self; *percentage* greater than 0.5: more money to Player 2).

In conditions 1 and 2, Player 2 was presented with the reward money already split (10 to Player 1, 5 to Player 2). Here, the dependent variable was the magnitude of change from the original split to the final split: $10 - x/15$, with $x$ being the final number of coins given to Player 1. In condition 3, Player 2 was presented with an unsplit reward and the dependent variable was calculated as for Player 1.

We decided to use the dictator game as a realistic, ecologically valid, measure of cooperation. In real-life interaction, it is common for people to be in cooperative situations where they have to share common resources. Then, attributing a larger share of the common resource to a partner is more often perceived as an other-oriented decision than a self-reward/self-punishment action [7,11,12]. To test the impact of other factors on the reward split, we ran general linear models investigating the effect of friendship index, guilt felt/judged guilt and condition on the reward split (see electronic supplementary material for details).

# 3. Results

## 3.1. Feeling guilty (Player 1)

### 3.1.1. Felt guilt and facial movements

After receiving the feedback in Stage 4, Player 1 experienced an increase in guilty feelings ($M = -0.66$, CI = [−0.90–0.43]; $t_{106} = -5.62$, $p < 0.001$) and shame feelings ($M = -0.53$, CI = [−0.72–0.34]; $t_{106} = -5.6$, $p < 0.001$). Players reported a significantly higher level of guilt than shame after the induction task ($M = -0.19$, CI = [−0.37–0.0005]; $t_{106} = -1.99$, $p = 0.049$), suggesting that guilt was the primary induced emotion (see electronic supplementary material, table S3 for details).

#### 3.1.1.1. Comparison of guilt and control conditions

The results of the bootstrap test, creating expected distributions for action units (AUs) based on the control condition and comparing those with the observed distribution of action units in the guilt condition, revealed that participants in the guilt condition exhibited facial muscle activation that was significantly different from the control condition. Electronic supplementary material, table S5 presents the summary of the comparison for the entire guilt dataset. In the upper face, AU4 was more active in the guilt condition more than in the control videos. In the lower face, AU10, AU12 and AU20 were active more often than would have been predicted based on the control videos. Most striking was the difference in the likelihoods of participants to touch their face, hair, ear, scratch or laugh in the guilt videos. Participants in the guilt videos were significantly less likely to show activation of AU24.

#### 3.1.1.2. Comparison of weak guilt and strong guilt samples

In the direct comparison between the two groups (individuals who expressed changes in feeling of guilt and those who did not; electronic supplementary material, table S6), individuals who reported strong feelings of guilt were more likely than expected to touch their ear and laugh (corroborating the results of comparing guilt videos with the control videos). AU1, AU2, AU4, AU14 and AU24 were significantly less likely to occur in participants who reported strong guilt.

### 3.1.2. Reward split

On the whole Player 1 gave a bigger share to Player 2, keeping an average of six coins (out of 15) for themselves ($M = 5.97$, s.d. $= 1.88$) and giving nine coins to their partner ($M = 9.03$, s.d. $= 1.88$; $t_{106} = -8.41$, $p < 0.001$). Player 1 gave more money to their partner the guiltier they felt ($\beta = 0.021$; s.e. $= 0.0096$; $p = 0.0302$). There was no effect of friendship on reward split ($\beta = 0.0040$; s.e. $= 0.0063$; $p = 0.523$) and whether Player 1 was blamed or not for his/her performance did not influence reward split ($\beta = 0.0033$; s.e. $= 024$; $p = 0.890$).

## 3.2. Perceiving guilt in others (Player 2)

### 3.2.1. Judged guilt and facial movements

**Control videos.** The control videos presented to judges that were judged as presenting guilt differ significantly from other control videos by showing more activity in AU5, AU18, AU24, face touching and scratching (electronic supplementary material, table S7). They also showed less activity of AU1, AU2, AU7 and AU20.

**Guilt videos.** The guilt videos that were judged as presenting guilt differ significantly from other guilt videos by showing more activity in AU18, AU24 and ear touching (electronic supplementary material, table S8). They also show less activity of AU10, AU12, AU14 and face touching.

We investigated which other factors influenced the judgement of guilt by Player 2: none of the investigated variables affected judged guilt (condition $\beta = -0.13$, s.e. $= 0.083$, $p = 0.123$; PoO $\beta = -0.062$, s.e. $= 0.079$, $p = 0.435$; friendship index $\beta = 0.032$, s.e. $= 0.076$, $p = 0.673$).

### 3.2.2. Reward split

On the whole, Player 2 did not favour themselves over their partner in the reward split, keeping seven coins (out of 15) on average for themselves ($M = 7.39$, s.d. $= 1.46$) and giving eight coins to their partner ($M = 7.69$, s.d. $= 1.44$; $t_{106} = -0.87$, $p = 0.387$). However, this does mean that Player 2 changed the original split presented to them, from 10 coins for their partner to 8 on average ($t_{106} = 13.52$, $p < 0.001$). In relationships with a strong friendship index (*both players reported a friendship index greater than or equal to 4*), Player 2 was more likely to change the original split (i.e. taking more money for themselves than the original five coins) when they judged their friends as seeming guilty (judged guilt greater than 3; $\beta = 0.081$; s.e. $= 0.031$; $p < 0.01$; see figure 3), attributing more money to themselves than to their partner. To assess the robustness of this model, we reran the analysis without any extreme values (i.e. outliers; any value greater than 1.5 s.d. from the mean, $n = 10$). This interaction did not hold with the removal of these data points; either potentially due to extreme values driving the model, or, due to the reduction in sample size. We, therefore, want to highlight that this finding should be approached with caution.

# 4. Discussion

In this experiment, we examined the social consequences of both feeling guilty and observing guilt in others. First, we examined the behavioural outcomes associated with guilt, and the facial movements co-occurring with self-reported increase in guilty feelings. Second, we examined the behavioural outcomes of being the victim of wrongdoing, and how this is affected by friendship. We found that participants who were told they were responsible for the failure of the cooperative game, allocated a greater proportion of the reward to their partner the more guilty they felt (regardless of the strength of their friendship). Similarly, players punished partners more when they looked guilty, and this effect was modulated by friendship: players punished their close friends more when they were perceived as feeling guilty.

When told they performed poorly, participants gave more money to their partners the guiltier they felt, regardless of their friendship, which is consistent with previous findings highlighting the positive social consequences associated with feelings of guilt [17–19,21]. This could function as both reputation and relationship repair, which are not always based on friendship but are inherent to human interactions [12]; acknowledging that the wrongdoing was committed, but also indicating that the wrongdoing was unintentional and further retaliation is not necessary [9,10]. The participants' personality traits affected some of the decisions made. Pro-social personality traits (agreeableness and emotional stability) correlated positively with the level of reported guilt. This supports the idea that guilt can be adaptive and has a positive social function [43–45]. Moreover, conscientious Player 1s

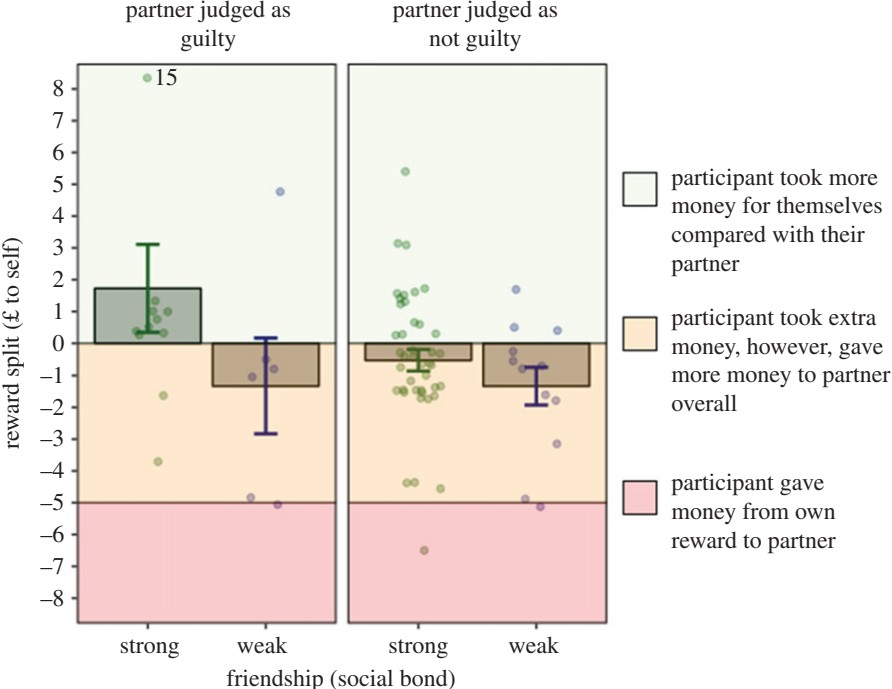

**Figure 3.** The interaction between judged guilt and friendship. Reward split is presented as a difference between the final amount taken by Player 2 and the amount attributed to Player 1. To better visualize the data for this figure, friends were split into 'weak friendship' and 'strong friendship' based on a median split: if friend index was higher than 4, participants were considered as close friends. Judged guilt was turned into a binary variable for this graph, using a median split: if judged guilt was higher than 3 on the PANAS, Player 2 judged Player 1 as guilty (guilty); if the judged guilt was lower than 3, Player 2 did not judged Player 1 as guilty (not guilty). The line at −5 represent the original split: 5 coins attributed to Player 2 and 10 coins attributed to Player 1. The line at 0 represents financial equality (which could not be obtained as we presented Player 2 with an odd number of coins). Anything above the middle line means Player 2 claimed more money for themselves.

gave a bigger share of the reward to Player 2. This personality trait is linked to diligence and could indicate a propensity to doing the right thing and making amends. The annoyance felt towards oneself also had an interesting effect on the decisions made by both players. On the one hand, Player 1 reported higher self-annoyance, felt guiltier and gave more money to Player 2. On the other hand, when Player 2 reported a higher level of self-annoyance, they judged Player 1 as looking guiltier than when they reported a low self-annoyance. Player 2 might have projected their own emotions onto Player 1, assuming the official wrongdoer would feel as bad as they did. Being blamed by the partner did not add to the existing guilt. This supports previous findings by Parkinson & Illingworth [46] of a slight tendency for blame to reduce guilt in the presence of self-blame.

When presented with an unfair reward split, the victim changed this unequal split to a fairer split (i.e. took more money for themselves than the original £5) and did so more when the wrongdoer was both a close friend and guilty. This resulted in an almost 50–50 split of the shared reward, which is atypical of the dictator game. However, we need to acknowledge the various differences between a 'typical' dictator game and the version used here. In most studies using a dictator game, one participant needs to divide a set reward between two people (either themselves and another player or two distinct people [47]). In our version, participants were presented with the reward already split. In 'typical' dictator games, how the participant with the role of the dictator (the one deciding how to split the reward) distributes the money varies immensely, but they tend to attribute about 42% of the share to their partner [48]. The interpretation of non-zero transfers has proven very complex; however, the main argument is that they represent a measure of social preference as well as people's heterogeneity [47,48]. An alternative view suggests that the reward split observed in our study could measure Player 2's politeness and need for fairness more than punishment of Player 1 [47]. It could also be an anchoring effect [49] of the original 70–30 original split Player 2 was presented with. An anchoring effect is a bias towards an arbitrary value provided before one needs to make numerical judgement [49]; in a dictator game, if the dictator is provided with the information that their partner worked twice as hard as they did, we might observe that dictator provide more money to their partner compared with dictators that were not told anything.

In our study, as the original split was in favour of Player 1, Player 2 might have not changed the split as much as if the original split was in their favour (equal split not possible in our setting).

Overall, this is in contrast with previous studies showing people being more lenient towards people expressing guilt [26,50,51]. One reason might be that our study examines the function of guilt between interacting participants, rather than via a hypothetical game [18,21] or autobiographical recall [52], making our experiment more ecologically valid than any previous research. Alternatively, it could be that participants evaluate the contribution of their partner differently in light of the guilty look. The guilty look might reinforce the information given to the participant that their partner has performed poorly and act as an acknowledgement of the wrongdoer's bad behaviour [13]. After receiving the feedback from the game, participants did not interact directly anymore. Therefore, any active expression of remorse might not have happened because the context for Player 1 to do so did not present itself. Player 1's decision regarding the reward split could be considered as an expression of remorse. Moreover, we need to consider the fact that most participants within a pair knew each other prior to the study. Even though not all participants reported a high level of friendship, the fact that they knew each other might help explain further the results observed in this study.

The punishment function of guilt was further enhanced when victims had a strong friendship with the wrongdoer suggesting that the victim punished their guilty friends more. This result could be due to at least two reasons. First, Player 2 might have acted differently when facing someone they perceived as guilty when that someone was a close friend compared with when it was an acquaintance or a stranger. Second, Player 2 might have perceived their close friends as expressing more guilt than an acquaintance or a stranger, so the nature of their relationship might have impacted Player 2's perception of Player 1's emotional state. People sharing a strong friendship bond have invested a lot of time and resources into the relationship; any disruption to the relationship ought to be repaired in order to save and restore the relationship [10], regardless of short-term consequences such as punishment. Recent research showed the importance of guilty apology in strong friendship [13]. Moreover, opportunities to make amends are more plentiful among friends than among strangers, which lead to the idea that in order to maintain a strong, honest relationship, punishment might sometimes be necessary [10,11,13]. Our finding supports the idea that guilt has a much stronger function and impact within existing friendships, leading to harsher punishments. This is in line with previous research reporting clearer emotional expressions among friends [22,23].

We used a bottom-up coding scheme to identify facial patterns associated with the experience (production and perception) of guilt. We found that players informed of their poor performance and experiencing guilt (Player 1) displayed more frown (AU4 brow lowerer), upper lip movement (AU10 upper lip raiser), smile (AU12 lip corner puller) and lip stretch (AU20, lip stretch [40]). They also presented more self-directed behaviours: ear touching, hair touching, face touching and scratching; and laughed more than when told how the cooperative game would work (control condition). Ear touching and laughter were even more present in participants reporting high level of guilt after induction. Interestingly, some of those movements were previously documented in association with guilt (AU4 and AU20 [24]). We also found that the identification of guilt in others was associated with ear touching and lip movements: pucking (AU18 lip pucker) and pressing (AU24 lip presser). Therefore, we found no direct correspondence between facial movements and perception of guilt, supporting current research that the link between facial expression and emotion could be less strong than previously thought [53]. Moreover, we did not find any impact of the face alone on the victim's response (condition 3), and our experimentally induced facial expressions did not have an impact (no difference between conditions 1 and 2). Finally, it is possible that the self-annoyance felt was expressed through specific facial movements and influenced the perception of guilt. Further research would be needed to disentangle the facial movements associated with guilt and those associated with self-annoyance specifically. Therefore, although we found that judgements of guilt have an impact on social outcomes, we do not know which behavioural (or other) factors were influencing these judgements.

It is important to remain cautious when interpreting these new findings. First, even though both participants reported variation in the amount of guilt they felt throughout the experiment, the reported values remained under the scale midpoint (2.5, using the 1–5 PANAS). This experiment was designed to induce ethically appropriate levels of guilt and mimic real-life situations. Our goal was, therefore, to induce mild levels of guilt and mostly to observe variations in the levels of guilt throughout the experiment. Second, caution should always be used when interpreting self-report measures; it is possible participants down-played how much guilt they were feelings or were not able to identify the feelings they experienced as the study went on. Third, the experimental induction of guilt might have had different outcomes based on participants' propensity to believe what they were told. If both players were

aware of the number of times the marble fell off the platform, some Player 1s might not have believed that they were more to blame for those falls. Fourth, another confounding factor that could explain Player 2's decision is having just taken part in a cooperative task with a partner; the face might not have influenced Player 2's final decision even when Player 2 reported that Player 1 'looked' more guilty (Stage 4—Player 2). It would be interesting to take this into account in a follow-up study, designing a task where participants are introduced but do not need to cooperate, as well as introducing a solitary task in order to disentangle possible confound factors present in our experiment.

## 5. Conclusion

This is the first study examining the impact of friendship on the consequences of guilt in a social interaction between participants. We examined genuine social interactions between dyads differing in relationship quality and tested the impact of guilt within the interaction. In line with previous research, we demonstrated the pro-social role of guilt: guilty players gave more money to their partners. We might have also revealed a new finding demonstrating a punishment effect of guilt, and a modulating effect of friendship on victims' behaviours: victims of a wrongdoing punished their close guilty friends more than less close guilty friends. It could, therefore, seem costly for the wrongdoer to communicate guilt; however, as punishment after wrongdoing has been demonstrated to benefit the relationship in the long term [9,13], the long-term benefits could outweigh this cost. More research is required into the effect of guilt on social interactions, comparing social and asocial situations as well as controlling for people's gullibility. Overall, the findings presented in this paper seem to indicate that the social function of emotions may differ depending on the quality and type of relationship between interacting partners.

Ethics. The projects have been reviewed and approved by the Science Faculty Ethics Committee (SFEC) of the University of Portsmouth (SFEC 2017-107). Written consent was obtained from all participants prior to participation.

Data accessibility. All data generated and analysed in this study are available at https://doi.org/10.5061/dryad.tmpg4f4vw [54]. Electronic supplementary material is available for this paper.

Authors' contributions. The first, fifth and second authors developed the study concept. Data collection was performed by the first and second authors. The first, second and fifth authors analysed and interpreted the data. The first, second and fifth authors drafted the manuscript and all authors provided critical revisions. All authors approved the final version of the manuscript for submission.

Competing interests. We declare we have no competing interests.

Funding. Those studies were funded by a Leverhulme Trust Research Project Grant *Cultural variation in the social function and expression of guilt* awarded to the fifth and third authors, and a European Research Council Consolidator Grant 864694 FACEDIFF to Bridget Waller.

Acknowledgement. We thank Peter Clark for help with reliability analysis. We thank Alexander Mielke for his advice and review. We also thank two anonymous reviewers for their constructive input.

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
