## [Reviewer comments · Royal Society Open Science]

Review History

RSOS-200617.R0 (Original submission)

Review form: Reviewer 1

Is the manuscript scientifically sound in its present form?

Yes

Are the interpretations and conclusions justified by the results?

Yes

Is the language acceptable?

Yes

Do you have any ethical concerns with this paper?

No

Have you any concerns about statistical analyses in this paper?

No

Recommendation?

Accept as is

Comments to the Author(s)

The experimental set-up strikes me as innovative and effective. And the results are interesting, and worth publishing.

Decision letter (RSOS-200617.R0)

Dear Dr Julle-Danière

The Editors assigned to your paper RSOS-200617 "The social function of the feeling and expression of guilt" have now received comments from reviewers and would like you to revise the paper in accordance with the reviewer comments and any comments from the Editors. Please note this decision does not guarantee eventual acceptance.

Please submit your revised manuscript and required files (see below) no later than 21 days from today's (ie 03-Sep-2020) date. Note: the ScholarOne system will 'lock' if submission of the revision is attempted 21 or more days after the deadline. If you do not think you will be able to meet this deadline please contact the editorial office immediately.

Kind regards,
Andrew Dunn

Royal Society Open Science Editorial Office
Royal Society Open Science
openscience@royalsociety.org

on behalf of Dr Narayanan Srinivasan (Associate Editor) and Essi Viding (Subject Editor)
openscience@royalsociety.org

Associate Editor Comments to Author (Dr Narayanan Srinivasan):

Associate Editor: 1

Comments to the Author:

One reviewer has commented on the paper and is positive.

I do have some concerns, which are given below and the authors are requested to address those concerns and submit the final version of the paper.

(1) Abstract makes no mention of the method used. While i know this style is used by some, i find it difficult to understand the result if i have no idea what method was used.

(2) At the end of introduction, authors mention the use of "dictator game". Perhaps a rationale for picking the dictator game can be explicitly provided given that other games are potentially available.

(3) Did the authors have any hypotheses regarding how friendship affects how wrongdoers who are perceived as guilty are treated? Was this study exploratory?

(4) It was not clear how the sample size was decided and what was the stopping criteria used, if any? Some information can be provided.

(5) It was not clear whether the order of the questionnaires before the subjects played the game was fixed. Supplementary material says order of questions was randomized but it seems to imply that this was within a questionnaire. The URCS seems to come always at the end just before people played the game. Would that affect anything?

(6) Page 10 - p = .049. See earlier comment on sample size estimation.

(7) Feeling of guilt and shame - Was there any difference between europeans and east asians? was it affected by whether they came as pairs or signed up alone. Majority of subjects seems to have come as pairs.

(8) Again, any cultural difference in terms of reward split (page 11)

(9) The rationale for median split was not clear to me. Why not use friendship strength as a continuous variable rather than arbitrarily dichotomizing it using median split? What was the distribution of friendship strength? Also it appears that there are outliers influencing the results. There is a subject (green dot, number 15 appears on the side in figure 3) in strong friendship group, who seems a definite outlier and probably is contributing to the effect. Please check whether this is an outlier (visually appears to be the case) and if so, analyze after dropping that subject. There could be other outliers.

(10) One issue that is not discussed very well in the discussion is the fact majority of the people are known to each other and that itself may be the reason for less leniency. Authors mention that "Participants were given the opportunity to either sign up with one of their friends or sign up alone and be paired up with a stranger (36 participants signed up alone to be paired up)." Perhaps this is one reason the results are different from other studies, which may have paired up strangers. Perhaps the authors can clarify or discuss this aspect.

Reviewer comments to Author:

Reviewer: 1

Comments to the Author(s)

The experimental set-up strikes me as innovative and effective. And the results are interesting, and worth publishing.

===PREPARING YOUR MANUSCRIPT===

a 'clean' version of the new manuscript that incorporates the changes made, but does not highlight them. This version will be used for typesetting if your manuscript is accepted. Please ensure that any equations included in the paper are editable text and not embedded images.

===PREPARING YOUR REVISION IN SCHOLARONE===

- Ensure that your data access statement meets the requirements at <https://royalsociety.org/journals/authors/author-guidelines/#data>. You should ensure that you cite the dataset in your reference list. If you have deposited data etc in the Dryad repository, please include both the 'For publication' link and 'For review' link at this stage.
- If you are requesting an article processing charge waiver, you must select the relevant waiver option (if requesting a discretionary waiver, the form should have been uploaded at Step 3 'File upload' above).
- If you have uploaded ESM files, please ensure you follow the guidance at <https://royalsociety.org/journals/authors/author-guidelines/#supplementary-material> to include a suitable title and informative caption. An example of appropriate titling and captioning may be found at https://figshare.com/articles/Table_S2_from_Is_there_a_trade-off_between_peak_performance_and_performance_breadth_across_temperatures_for_aerobic_scope_in_teleost_fishes_/3843624.

Author's Response to Decision Letter for (RSOS-200617.R0)

See Appendix A.

Decision letter (RSOS-200617.R1)

Dear Dr Julle-Danière,

It is a pleasure to accept your manuscript entitled "The social function of the feeling and expression of guilt" in its current form for publication in Royal Society Open Science. The comments of the reviewer(s) who reviewed your manuscript are included at the foot of this letter.

on behalf of Dr Narayanan Srinivasan (Associate Editor) and Essi Viding (Subject Editor)
openscience@royalsociety.org

Associate Editor Comments to Author (Dr Narayanan Srinivasan):

The authors have done their best to address the comments and the paper is now suitable for publication.

Appendix A

We would like to thank you for the thorough reviews of this paper. Here we will address each comment (blue) sequentially (our response in black). When referring to a change in the paper the line numbers are given in bold and refer to the tracked changes version of the manuscripts.

Associate Editor Comments to Author (Dr Narayanan Srinivasan)

I do have some concerns, which are given below and the authors are requested to address those concerns and submit the final version of the paper.

1. Abstract makes no mention of the method used. While i know this style is used by some, i find it difficult to understand the result if i have no idea what method was used.

Thank you, we added a sentence regarding the Method used **II.20-23**.

2. At the end of introduction, authors mention the use of "dictator game". Perhaps a rationale for picking the dictator game can be explicitly provided given that other games are potentially available.

A rationale for using the dictator game is now presented in the 'Measures' section (II.184-189) but we thank you for your comment and have added a short explanation for choosing this game in the introduction (**II.86-89**).

3. Did the authors have any hypotheses regarding how friendship affects how wrongdoers who are perceived as guilty are treated? Was this study exploratory?

This study was indeed exploratory and as such, we did not have hypothesis regarding the effect of friendship. Based on previous studies, we could have expected guilty wrongdoers to be treated more leniently, but as friendship had never been investigated in this context, we did not formulate hypotheses.

4. It was not clear how the sample size was decided and what was the stopping criteria used, if any? Some information can be provided.

Thank you for pointing this out. The sample size for this study was based on previous research looking at the induction of guilt. We could not run a power analysis prior to the study as we did not know the effect of the factors investigated and could not use previous research to estimate those. Moreover, due to financial limitation, and to keep the design as simple as possible, we did not include a 'no guilt' condition for Player 1. This allowed us to minimise the number of variables, already extensive, and reduce the number of participants needed. Therefore, in order to capture as many facial expressions as we could in a culturally varied sample, we aimed to have 50 to 60 participants posing as Player 1 in each condition (2 conditions for Player 1: 'blame' vs. 'no blame'). As a result, we aimed to collect data from 110 pairs of participants. This information has been added to the manuscript (**II. 97-103**).

5. It was not clear whether the order of the questionnaires before the subjects played the game was fixed. Supplementary material says order of questions was randomized but it seems to imply that this was within a questionnaire. The URCS seems to come always at the end just before people played the game. Would that affect anything?

Thank you for your comment, this has been clarified in the Sup. Mat. **II.35-36**. The order of questionnaires was indeed fixed within blocks throughout the experiment. We did not expect the order presentation to influence participants responses or reactions to the game, but if the order did impact the participants, the impact would be the same for all. As participants were only tested once, we did not consider this to be an issue for our experiment.

6. Page 9 - $p = .049$. See earlier comment on sample size estimation.

This p value is very close to non-significance and we try to acknowledge that and exercise caution throughout. As this is an exploratory study, more research is needed to help understand the effect of the guilt induction on Player 1's feelings.

7. Feeling of guilt and shame - Was there any difference between Europeans and East Asians? Was it affected by whether they came as pairs or signed up alone. Majority of subjects seems to have come as pairs.

8. Again, any cultural difference in terms of reward split (page 10)

Thank you for your comment. Culture was not a factor of interest in this study. Our sample included participants that self-identified as belonging to either a European or an (East-)Asian ethnicity, but we emphasise this as a non-WEIRD sample and thus more representative. We were interested in holistic results, but we acknowledge that it is a very interesting question that would require further research. Most participants did sign up as pairs, but the closeness/friendship questionnaire revealed that this was not an indicator of objective closeness. Indeed, some participants that came as a pair met their partner a few weeks prior to the study (first year students starting University late September, study running from December to April) and did not feel particularly close to them. Therefore, we considered the friendship index a more objective factor to investigate rather than splitting our sample between pairs that signed up together and pairs that were randomly formed. The level of friendship did not affect the self-reported feelings (Sup Mat II. 187-188).

9. The rationale for median split was not clear to me. Why not use friendship strength as a continuous variable rather than arbitrarily dichotomizing it using median split? What was the distribution of friendship strength?

Thank you for your comment, we hope that the following explanation will clarify this point. Despite not being able to split our sample by pair type (see above), our recruitment practice still resulted in a bimodal distribution: participants were invited to either sign up with a friend or alone. Therefore, as shown in the histogram below, the friendship index distribution was bimodal: many people are not friends, and many are strong friends (5-7), but fewer people are in between. We could have just trusted the fact that 'friends' signed up together, but we wanted to double check using this median split method.

Histogram with normal curve

The friendship index was controlled for in the bootstrapping analysis and the median split was used to ensure that friends and non-friends were distributed equally when testing other conditions, as explained in the Supplementary Materials II. 125-128: “The bootstraps were conducted so that the ratio of the two places of origin and the two friendship categories (weak friendship: *if one or both players reported a friendship index $4 <$* ; strong friendship: *both players reported a friendship index ≥ 4*) in the null condition were identical with the ratio in the test condition, to avoid results being driven by different underlying group structures.”. This did not impact the results of the bootstrapping analysis in any way but was a necessary step to ensure that there was no systematic difference in group structure between the null condition and our datasets.

For consistency purposes, we used the bimodal variable in all the analyses conducted (bootstrap and GLM).

Also it appears that there are outliers influencing the results. There is a subject (green dot, number 15 appears on the side in figure 3) in strong friendship group, who seems a definite outlier and probably is contributing to the effect. Please check whether this is an outlier (visually appears to be the case) and if so, analyze after dropping that subject. There could be other outliers.

We agree that some extreme values could be influencing the fit of our models. In addition to the ‘number 15’ datapoint, 9 more datapoints could be considered extremes (or, outliers), appearing more than 1.5 standard deviations from our mean. However, we are hesitant to remove these from our dataset and analysis for several reasons. These datapoints represent authentic decisions made by our participants (as opposed to an error in measurement) and reflect legitimate responses made by our participants in our scenario. GLM’s are quite robust to non-normal distributions, and therefore we do not believe we have justification to remove these points entirely. Our analysis is also sensitive to smaller sample sizes. We admit our sample-size is already on the lower end, and therefore if we removed these extremes that would account for more than 10% of our data, which could make our findings less (rather than more) robust.

We do, however, agree that if there are any datapoints that are significantly influencing the fit of our models, we should then approach our findings with much more caution. If we refit our data excluding the ‘number 15’ datapoint, or, by removing all values more than 1.5SD, the interaction between Friendship and Judged Guilt does not hold ($\beta = 0.03$; $SE = 0.025$, $p < 0.15$ and $\beta = 0.02$; $SE = 0.014$, $p < 0.14$ respectively). We do not believe this invalidates our original findings, but this does suggest that our models are not overly robust. We now discuss this transparently in our results, and emphasise the added caution we should take because of this (II. 249-253) :

“To assess the robustness of this model, we reran the analysis without any extreme values (i.e, outliers; any value $>1.5SD$ from the mean, $n=10$). This interaction did not hold with the removal of these data points; either potentially due to extreme values driving the model, or, due to the reduction in sample size. We therefore want to highlight this finding should be approached with caution.”

10. One issue that is not discussed very well in the discussion is the fact majority of the people are known to each other and that itself may be the reason for less leniency. Authors mention that "Participants were given the opportunity to either sign up with one of their friends or sign up alone and be paired up with a stranger (36 participants signed up alone to be paired up)." Perhaps this is one reason the results are different from other studies, which may have paired up strangers. Perhaps the authors can clarify or discuss this aspect.

Thank you, we have included this point in the discussion (**II.334-337**).